# Waist circumference and low high-density lipoprotein cholesterol as markers of cardiometabolic risk in Kenyan adults

Daniel Faurholt-Jepsen[1], Henrik Friis[2], David L. Mwaniki[3], Michael K. Boit[4], Lydia U. Kaduka[3], Inge Tetens[2], Dirk L. Christensen[5]*

1 Department of Infectious Diseases, Copenhagen University Hospital (Rigshospitalet), Copenhagen, Denmark, 2 Department of Nutrition, Exercise and Sports, University of Copenhagen, Copenhagen, Denmark, 3 Centre for Public Health Research, KEMRI, Nairobi, Kenya, 4 Department of recreation Management and Exercise Science, Kenyatta University, Nairobi, Kenya, 5 Department of Public Health, Section of Global Health, University of Copenhagen, Copenhagen, Denmark

* dirklc@sund.ku.dk

## Abstract

### Background

Abdominal obesity predict metabolic syndrome parameters at low levels of waist circumference (WC) in Africans. At the same time, the African lipid profile phenotype of low high-density lipoprotein (HDL) cholesterol without concomitant elevated triglyceride levels renders high triglyceride levels detrimental to cardiometabolic health unsuitable for identifying cardiometabolic risk in black African populations.

### Objectives

We aimed to identify simple clinical measures for cardiometabolic risk based on WC and HDL in an adult Kenyan population in order to determine which of the two predictors had the strongest impact.

### Methods

We used linear regression analyses to assess the association between the two exposure variables WC and HDL with cardiometabolic risk factors including ultrasound-derived visceral (VAT) and subcutaneous adipose tissue (SAT) accumulation, fasting and 2-h venous glucose, fasting insulin, fasting lipid profile, and blood pressure in adult Kenyans (n = 1 370), and a sub-population with hyperglycaemia (diabetes and pre-diabetes) (n = 196). The same analyses were performed with an interaction between WC and HDL to address potential effect modification. Ultrasound-based, semi-quantitative hepatic steatosis assessment was used as a high-risk measure of cardiometabolic disease.

### Results

Mean age was 38.2 (SD 10.7) (range 17–68) years, mean body mass index was 22.3 (SD 4.5) (range 13.0–44.8) kg/m$^2$, and 57.8% were women. Cardiometabolic risk was found in

**Data Availability Statement:** Data cannot be shared publicly without permission from the Kenyan health authorities (KEMRI). In order to request for data access, contact: Kebenei Enock

Kipchirchir Acting Head of the Scientific and Ethics Review Unit (SERU) kisacheienock@gmail.com.

**Funding:** DLC: (J. no. 104.DAN.8-871, RUF project no. 91202); Cluster of International Health, University of Copenhagen (no grant no); Beckett Foundation (no grant no); Dagmar Marshall Foundation (no grant no); Dr. Thorvald Madsen's Grant (no grant no); Kong Christian den Tiende's Foundation (no grant no); Brdr. Hartmann Foundation (no grant no) The funders had no role in study design, data collection and analysis, decision to publish, or preparation of the manuscript.

**Competing interests:** The authors have declared that no competing interests exist.

the association between both WC and HDL and all outcome variables (p<0.05) except for HDL and SAT, fasting and 2-h venous glucose. Additive cardiometabolic risk (WC and HDL interaction) was found for SAT, low-density lipoprotein cholesterol, and triglycerides. No differences in the association between WC and HDL and the outcome variables were found when comparing the full study population and the hyperglycaemia sub-population. Increase in WC and HDL were both associated with hepatic steatosis (OR 1.09, p<0.001, and OR 0.46, p = 0.031, respectively).

## Conclusion

In adult Kenyans, increasing WC identified more cardiometabolic risk factors compared to HDL.

## Introduction

The concept of the hypertriglyceridemic waist phenotype and the potential of using a combination of simple clinical measures (waist circumference (WC) and plasma triglyceride) as a marker of elevated cardiovascular risk was introduced by Després and co-workers in 2000 [1, 2]. An African diaspora population was included in one of the studies [1], and the authors suggested that for a given WC, this population compared to a North American white population had a generally more cardio-protective plasma lipoprotein profile, including lower plasma triglyceride levels, due to lower visceral adipose tissue (VAT) accumulation and higher plasma lipoprotein lipase activity. Low triglyceride levels have also been shown in studies carried out in black South Africans [3, 4] as well as in rural Kenyans [5]. In the latter, we showed that the prevalence of dyslipidaemia was high (~37%) in both men and women, and that almost 9 in 10 had isolated hypoalphalipoproteinemia, i.e. low high-density lipoprotein cholesterol (HDL) dyslipidaemia. Furthermore, Delisle and co-workers reported low HDL across body mass index groups including underweight individuals in Benin [6]. It is of note that Després and co-workers used plasma triglyceride levels of ≥2.0 mmol/L and WC values of ≥90 cm as cut-offs for elevated levels [1]. Neither of the two cut-offs match the black African lipid or body composition profile phenotype; few exceed the 1.7 mmol/L cut-off for dyslipidaemia [5, 7], and a recent meta-analysis in a Pan African population (~25 000 individuals) on finding WC cut-offs for having at least two metabolic syndrome traits showed that cut-offs were similar in men and women at 81.2 and 81.0 cm, respectively [8].

Thus, replacing triglyceride with HDL in combination with lower WC cut-off than used by Després and colleagues seems more appropriate as cardiometabolic risk markers in black, African populations.

The aim of this study was to identify simple clinical outcome variables of cardiometabolic risk in adult Africans, including similar analyses in a sub-group of people with hyperglycaemia, based on body composition (WC) and lipid (HDL) phenotypes suitable for black African populations.

## Methods

A sample of 1 449 adult rural and urban Kenyans participated in a study on cardiometabolic risk factors [5, 9]. In brief, the study population was based on a convenience sample, even though random sampling in the rural area and sampling of urban biological family members

of the rural participants was attempted, but failed. The rural population consisted of agro-fishing (Luo), agriculture (Kamba), and agro-pastoralist (Maasai) people, while the urban population consisted of the aforementioned ethnic groups as well as culturally related ethnic groups. All participants were informed about the study in writing as well as orally. Informed consent was signed or given as thumb print in case of illiteracy. Exclusion from this particular substudy was due to missing data on WC and HDL (n = 79). None were on lipid-lowering therapy, while 22 individuals were on oral hypoglycaemic agents or insulin therapy. Thus, 1 370 individuals (1 117 rural, 253 urban) were included in the study. Following an overnight (>8-h) fast, standard anthropometric measurements were carried out of which WC was measured with a measuring tape midway between the iliac crest and the costal margin following a quiet expiration. Ultrasound scanning of abdominal fat distribution, i.e. visceral (VAT) and subcutaneous (SAT) adipose tissue was measured using a standardised protocol [10]. Fasting plasma lipids were collected. Enzymatic colorimetric tests using the GPO-PAP [11], and the CHOD-PAP methods [12] were used to measure plasma triglycerides (TG) and total cholesterol (TC), respectively. The analysis was done using a Hitachi 912 System (Roche Diagnostics GmbH, Mannheim, Germany). A homogeny enzymatic colorimetric test was used for measuring plasma HDL-C, with HDL-C plus 2nd generation without pre-treatment being applied using a Hitachi 912 System (Roche Diagnostics GmbH, Mannheim, Germany). Plasma very low-density lipoprotein (VLDL) concentration was calculated according to the following equation [13]: VLDL = TG/2.2, while plasma low-density lipoprotein cholesterol (LDL) concentration was calculated as: LDL = TC—VLDL—HDL [13]. Venous whole blood glucose was analysed according to the blood glucose dehydrogenase method using haemolysation and deproteinisation using a B-HemoCue 201+ device (HemoCue AB, Ängelholm, Sweden). Subsequently, a 75-g oral glucose tolerance test was performed to determine 2-h glucose levels. Serum insulin was measured by a 1235 AutoDELFIA automatic immunoassay system (sensitivity 0.5 lU/ml) using time-resolved fluoro-immunoassay technique (kit no. BO80-101, PerkinElmer Life and Analytical Sciences, Wallac Oy, Turku, Finland). Systolic and diastolic blood pressures were measured twice on the right upper arm using a full-automatic device (Omron M6, HEM-7001-E, Kyoto, Japan), while the participant was seated. Central obesity was defined as WC ≥81.2 cm for men and 81.0 cm for women according to Ekoru and co-workers cut-offs [8], and low HDL values were defined as <1.0 mmol/L for men and <1.3 mmol/L for women [7]. Glucose tolerance status was classified according to World Health Organization/International Diabetes Federation criteria [14]. In a sub-group (n = 756), liver fat accumulation was assessed using ultrasound liver scans. This method is semi-quantitative and allows to distinguish between normal liver (score ≤ 4), mild (score between 5 and 7), moderate (score between 8 and 10) and severe (score ≥ 11) steatosis according to standardised criteria. For detailed methodological description, see [15]. Ethical approval was given by the National Ethical Review Committee in Kenya (SSC Protocol No. 936), and consultative approval was given by the Danish National Committee on Biomedical Research Ethics.

## Statistics

Descriptive data are presented as mean (SD) if normally distributed and as median (IQR) if skewed. Associations of WC and HDL were tested against cardiometabolic risk factors in unadjusted linear regression with WC and HDL as continuous and dichotomous variables and with interaction terms on WC and HDL. Furthermore, combined groups of low and high WC and HDL were tested against cardiometabolic risk factors using age and sex adjusted linear regression with combined low WC and high HDL as reference group. Skewed variables were log-transformed, thus the back-transformed coefficient $e^B$ should be interpreted as a ratio. P-

values <0.05 were considered statistically significant. All analyses were carried out using Stata 14.2 (IC version, Stata, College Station, USA).

## Results

The study population had a mean age of 38.2 (SD 10.7) years with 57.8% being women. Characteristics of the study population on anthropometry and body composition, biochemistry, and blood pressure are presented in Table 1. In age and sex adjusted linear regression analyses, HDL was correlated with VAT, LDL, TC, triglycerides, systolic and diastolic blood pressure, and insulin, while WC was correlated with all assessed cardiometabolic risk factors. Additive cardio-metabolic risk HDL and WC interaction was found for SAT, LDL, and triglycerides. For details, see Table 2. There was no difference in estimates in unadjusted analyses.

Both high vs. low WC and low vs. high HDL were individually correlated with all assessed cardio-metabolic risk factors (Table 3). The combined high WC and low HDL group had higher VAT, fasting and 2-h blood glucose levels compared to the combined low WC and high HDL, combined low WC and low HDL, as well as the combined high WC and high HDL groups (Table 4). The combined high WC and low HDL group had higher SAT, fasting insulin and systolic as well as diastolic blood pressure compared to the combined low WC and high

**Table 1. Characteristics of adult Kenyans (n = 1 370).**

|  | Mean/n(%) | SD |
|---|---|---|
| Age (years) | 38.2 | 10.7 |
| Female sex | 792 (57.8) |  |
| *Anthropometry and body composition* |  |  |
| Body mass index (kg/m$^2$) | 22.3 | 4.5 |
| Waist circumference (cm) | 79.5 | 11.4 |
| Visceral adipose tissue (cm) | 5.9 | 1.7 |
| Subcutaneous adipose tissue (cm) | 1.5 | 1.2 |
| Hepatic steatosis, n (%)* | 116 (15.3%) |  |
| Mild hepatic steatosis, n (%) | 108 (93.1%) |  |
| Moderate hepatic steatosis, n (%) | 8 (6.9%) |  |
| Severe hepatic steatosis, n (%) | 0 (0.0%) |  |
| *Biochemistry* |  |  |
| Fasting venous glucose (mmol/L) | 4.6 | 1.5 |
| 2-h blood venous glucose (mmol/L) | 5.4 | 2.6 |
| Fasting serum insulin (pmol/L)** | 23 | 15;36 |
| Fasting plasma total Cholesterol (mmol/L) | 3.9 | 1.0 |
| Fasting plasma HDL (mmol/L) | 1.1 | 0.3 |
| Fasting plasma total cholesterol/HDL ratio** | 3.5 | 2.9;4.4 |
| Fasting plasma LDL (mmol/L) | 2.3 | 0.8 |
| Fasting plasma VLDL (mmol/L) | 0.5 | 0.2 |
| Fasting plasma triglyceride (mmol/L)** | 0.9 | 0.7;1.2 |
| *Blood pressure (mmHg)* |  |  |
| Systolic | 120 | 16 |
| Diastolic | 74 | 10 |

Abbreviations: HDL: high-density lipoprotein cholesterol, LDL: low-density lipoprotein cholesterol, VLDL: very low-density lipoprotein cholesterol, SD: standard deviation.

*Based on ultrasound scanning semi-quantitative liver fat score (n = 756).

**Denoting median (interquartile range).

**Table 2. Association between high-density lipoprotein cholesterol (HDL) and waist circumference with body composition, biochemistry, and blood pressure in adult Kenyans (n = 1 370).**

| | HDL | Waist circumference | HDL * Waist circumference |
|---|---|---|---|
| Dependent variable | B (95% CI) | B (95% CI) | p for interaction |
| Visceral adipose tissue (cm) | -0.8 (-1.0; -0.5) ** | 0.1 (0.1; 0.1) ** | 0.903 |
| Subcutaneous adipose tissue (cm) | 0.1 (-0.1; 0.2) | 0.1 (0.1; 0.1) ** | 0.018 |
| Fasting venous glucose (mmol/L) | -0.2 (-0.4; 0.04) | 0.02 (0.01; 0.03) ** | 0.880 |
| 2-h venous glucose (mmol/L) | -0.4 (-0.8; 0.01) | 0.04 (0.03; 0.1) ** | 0.286 |
| Low-density lipoprotein cholesterol (mmol/L) | 0.3 (0.2; 0.4) ** | 0.02 (0.02; 0.03) ** | 0.004 |
| Total cholesterol (mmol/L) | 1.1 (1.0; 1.3) ** | 0.03 (0.02; 0.03) ** | 0.147 |
| Systolic blood pressure (mmHg) | 4.6 (2.2; 7.0) ** | 0.4 (0.4; 0.5) ** | 0.134 |
| Diastolic blood pressure (mmHg) | 2.0 (0.3; 3.6) ** | 0.3 (0.2; 0.3) ** | 0.312 |
| Dependent variable | $e^B$ (95% CI) | $e^B$ (95% CI) | p for interaction |
| Triglyceride (mmol/L)* | 0.75 (0.71; 0.81) ** | 1.01 (1.01; 1.01) ** | 0.008 |
| Fasting serum insulin (pmol/L)* | 0.89 (0.80; 1.00) ** | 1.02 (1.02; 1.02) ** | 0.945 |

Data are linear regression analyses adjusted for age and sex.

*Log-transformed.

**denotes p<0.05.

HDL, and the combined low WC and low HDL only. LDL and total cholesterol were highest in the high WC and high HDL group.

In a sub-group with hyperglycaemia (n = 196), mean age of 41.8 (SD 10.8) years, and with 63.8% being women (background characteristics shown in S1 Table), the same analyses showed similar results except for lack of correlation between HDL with DBP and insulin. No WC-HDL interaction was found in the sub-group analyses (S2 Table).

We tested for confounding between WC and HDL in mutually adjusted models, but found no change in effect size after adjustment. As for VAT accumulation comparing the group with normoglycaemia vs. the hyperglycaemia group, no WC-HDL interaction was found, and this

**Table 3. Association between low high-density lipoprotein cholesterol (HDL) and high waist circumference with body composition, biochemistry, and blood pressure in adult Kenyans (n = 1 370).**

| | HDL Low vs. high | Waist circumference High vs. low | HDL * Waist circumference |
|---|---|---|---|
| Dependent variable | B (95% CI) | B (95% CI) | p for interaction |
| Visceral adipose tissue (cm) | 0.3 (0.1; 0.5) ** | 1.7 (1.5; 1.9) ** | 0.296 |
| Subcutaneous adipose tissue (cm) | 0.4 (0.2; 0.5) ** | 1.6 (1.5; 1.7) ** | 0.012 |
| Fasting venous glucose (mmol/L) | 0.3 (0.1; 0.4) ** | 0.3 (0.2; 0.5) ** | 0.850 |
| 2-h venous glucose (mmol/L) | 0.6 (0.3; 0.8) ** | 0.9 (0.6; 1.2) ** | 0.865 |
| Low-density lipoprotein cholesterol (mmol/L) | -0.2 (-0.2; -0.1) ** | 0.6 (0.5; 0.7) ** | 0.099 |
| Total cholesterol (mmol/L) | -0.5 (-0.7; -0.4) ** | 0.7 (0.6; 0.8) ** | 0.557 |
| Systolic blood pressure (mmHg) | -2.8 (-4.5; -1.1) ** | 9.3 (7.6; 11.0) ** | 0.266 |
| Diastolic blood pressure (mmHg) | -1.1 (-2.7; 0.002) ** | 6.1 (5.0; 7.3) ** | 0.755 |
| Dependent variable | $e^B$ (95% CI) | $e^B$ (95% CI) | p for interaction |
| Triglyceride (mmol/L)* | 1.11 (1.06; 1.16) ** | 1.33 (1.27; 1.39) ** | 0.556 |
| Fasting serum insulin (pmol/L)* | 1.18 (1.09; 1.27) ** | 1.69 (1.57; 1.82) ** | 0.703 |

Data are univariate linear regression analyses.

*Log-transformed.

**denotes p<0.05.

**Table 4. Association between combined low/high waist circumference and high/low high-density lipoprotein cholesterol (HDL) with body composition, biochemistry, and blood pressure by adjusted means (95% CI) in adult Kenyans (n = 1 370).**

| Dependent parameter | Low WC and high HDL (reference group) | Low WC and low HDL | High WC and high HDL | High WC and low HDL |
|---|---|---|---|---|
| N (%) | 399 (29.1) | 525 (38.3) | 139 (10.2) | 307 (22.4) |
| Visceral adipose tissue (cm) | 5.1 (5.0; 5.3) | 5.5 (5.4; 5.7)[a] | 6.6 (6.4; 6.9)[a,b] | 7.2 (7.0; 7.4)[a,b,c] |
| Subcutaneous adipose tissue (cm) | 1.1 (1.0; 1.2) | 0.9 (0.9; 1.0)[a] | 2.5 (2.3; 2.6)[a,b] | 2.6 (2.5; 2.7)[a,b] |
| Fasting venous glucose (mmol/L) | 4.4 (4.2; 4.5) | 4.6 (4.4; 4.7)[a] | 4.6 (4.3; 4.9) | 4.9 (4.7; 5.0)[a,b] |
| 2-h venous glucose (mmol/L) | 5.0 (4.7; 5.2) | 5.3 (5.1; 5.5)[a] | 5.6 (5.2; 6.1)[a] | 6.0 (5.7; 6.3)[a,b] |
| Low-density lipoprotein cholesterol (mmol/L) | 2.3 (2.2; 2.3) | 2.1 (2.0; 2.2)[a] | 2.9 (2.8; 3.0)[a,b] | 2.6 (2.5; 2.7)[a,b,c] |
| Total cholesterol (mmol/L) | 4.1 (4.0; 4.1) | 3.4 (3.3; 3.5)[a] | 4.8 (4.6; 4.9) [a,b] | 4,1 (4.0; 4.2) [b,c] |
| Systolic blood pressure (mmHg) | 118 (117; 120) | 116 (114; 117)[a] | 126 (123; 128)[a,b] | 125 (124; 127)[a,b] |
| Diastolic blood pressure (mmHg) | 73 (72; 74) | 72 (71; 72) | 78 (77; 80)[a,b] | 77 (76; 79)[a,b] |
| Triglyceride (mmol/L)* | 0.8 (0.8; 0.8) | 0.9 (0.9; 0.9)[a] | 1.0 (0.9; 1.0)[a] | 1.1 (1.1; 1.2)[a,b,c] |
| Fasting serum-insulin (pmol/L)* | 18.7 (17.5; 19.9) | 18.9 (17.8; 19.9) | 33.7 (30.2; 37.5)[a,b] | 35.0 (32.6; 37.6)[a,b] |

Data are sex- and age adjusted means (95% confidence interval) with low WC and high HDL as reference group.

[a]Denotes significant difference from low WC/high HDL group

[b]Denotes significant difference from low WC/low HDL group

[c]Denotes significant difference from high WC/high HDL group

*Back-transformed adjusted means.

was sustained when repeating the analysis without the 22 participants on anti-hyperglycaemia medication.

WC and HDL increase were significantly associated with hepatic steatosis, OR = 1.09 (95% CI, 1.07; 1.11, p<0.001), and OR = 0.46 (95% CI, 0.23; 0.93, p = 0.031), and for every unit increase in WC and HDL, respectively. In the sub-group with hyperglycaemia, 90 participants were ultrasound scanned for liver steatosis assessment, and 30 (33.3%) had hepatic steatosis. In these, OR for every unit increase in WC was 1.10 (95% CI, 1.05; 1.16, p<0.001), and OR for unit increase in HDL was 0.38 (95% CI, 0.09; 1.60, p = 0.188). We identified 26 individuals with self-reported alcohol consumption that exceeded criteria for AFLD [16]. Excluding these individuals from the hepatic steatosis analyses did not alter the results.

## Discussion

In light of the growing evidence of a black African phenotype when it comes to cholesterol levels and central obesity, this study attempted to investigate whether central obesity measured as WC or HDL was the main "predictor" of an adverse cardiometabolic profile consisting of high values of ultrasound-measured VAT, simple biochemistry measures and blood pressure in adult, black Africans with hyperglycaemia. Hepatic steatosis was used to substantiate cardiometabolic risk assessment. In order to determine whether results differed in a cardiometabolic disease high risk group, we did the same analyses in a sub-group with hyperglycaemia.

In contrast to WC, HDL was not significantly correlated with all anthropometric, biochemical and blood pressure variables included in this study, suggesting that central obesity identifies more predictors for cardiometabolic risk. An interaction between WC and HDL on cardiometabolic risk was only found in three outcome variables, of which two were LDL and triglyceride. As their values are closely inter-correlated with HDL, the effect of combined HDL-WC measurements on cardiometabolic risk is of limited, clinical value.

Overall, in the high/low dichotomous and combined measurements, the strongest association with a less favourable cardiometabolic profile was found in the high WC and low HDL group. VAT was highest in the high WC and low HDL group, which is of concern as it may signify an atherosclerotic profile beyond the simple clinical data presented here. Although both high WC and low HDL were individually associated with determinants of cardiometabolic risk, we did not see any change in risk, when the two measures were combined in the analyses with mutual adjustment. Thus, in this context there was no confounding between central obesity and HDL on cardiometabolic risk.

Sam and colleagues have shown that people with type 2 diabetes and combined high WC and high TG had higher VAT accumulation and coronary artery calcium compared to groups with less adverse WC-TG combinations [17]. Our study supported these findings as far as VAT accumulation is concerned. Low HDL (replacing high TG) or high WC were both associated with VAT whether in the total study population or in the sub-group with hyperglycaemia. As we were not able to assess coronary artery calcium accumulation by computed tomography technique, or even carotid artery stiffness by ultrasound scanning technique for a broader cardiometabolic risk assessment, including stroke, we used assessment of hepatic steatosis as an additional risk factor for cardiometabolic disease. There is evidence that individuals with hepatic steatosis are at high risk of cardiovascular disease [18], and our results showed that WC increase was significantly associated with hepatic steatosis whether in the main study population or in the hyperglycaemia sub-group. This was only the case for HDL increase and hepatic steatosis in the main study population.

It is of note that standard plasma measurement of HDL may not capture the full range of HDL effects; therefore, particle concentrations of especially low and intermediate HDL subclasses may be better predictors of the anti-atherogenic properties of HDL mediated through (intra-abdominal) obesity [19]. In this study, when comparing the dichotomous and high/low combined measurements, we used different Pan-African cut-offs for low HDL compared to a previous study on dyslipidaemia in the same population (1.0 vs. 0.9 mmol/L for men and 1.3 vs. 1.0 mmol/L for women) [20], which by inference categorised a larger proportion of the current study population with hypoalphalipoproteinemia.

In regards to specific disease outcomes due to cardiometabolic risk, ischaemic heart disease remains relatively uncommon in sub-Saharan African populations as shown in a comprehensive review report by Onen [21], while increased central obesity and low HDL remains the dominant risk factors for stroke, and especially ischaemic stroke in these populations.

We acknowledge several limitations to this study. First, we did not test for hepatitis B or C which is a limitation to our hepatic measurements. Another limitation concerning the hepatic steatosis results we need to emphasize is the relatively low proportion (15.3% of which 93.1% was mild steatosis, and none had severe steatosis) we found in a sub-group of the study population. This could partly be explained by the low triglyceride levels seen in African populations, and such an association has been reported in African Americans compared to white and Hispanic US populations [22]. Furthermore, by focusing on WC and HDL and thus indirectly insulin resistance, we do not consider those at risk of diabetes due to pancreatic beta-cell failure as the predominant mechanism for dysglycaemia. Lastly, we acknowledge that the study population may not be representative of all Kenyans even though we did sample according to different and common dietary practices as well as in rural and urban areas.

In conclusion, our data suggest that measuring WC better predicts cardiometabolic risk factors compared to using HDL as a cardiometabolic exposure variable. The trend was similar whether based on results in a large study group including low-to-high cardiometabolic risk individuals, or in a high risk sub-group of individuals with hyperglycaemia. Significant associations with hepatic steatosis, a high-risk measure for cardiometabolic disease, and WC

whether in the general or in the hyperglycaemia study groups further substantiated WC as a better measure for cardiometabolic disease risk than HDL. Furthermore, combined WC-HDL sustained the conclusion of WC better predicting cardiometabolic risk factors.

## Supporting information

**S1 Table. Characteristics of adult Kenyans with impaired glucose tolerance (n = 196).**
(DOCX)

**S2 Table. Association between high-density lipoprotein cholesterol (HDL) and waist circumference with body composition, biochemistry, and blood pressure in adult Kenyans with impaired glucose tolerance (n = 196).**
(DOCX)

## Acknowledgments

We thank all participants, local chiefs, councils, politicians, and research teams responsible for data generation. Special thanks go to Professor Knut Borch-Johnsen, Copenhagen University Hospital (Holbaek, Denmark) for his invaluable contribution to the Kenya Diabetes Study in general. We acknowledge the permission by the Director of KEMRI to publish this manuscript.

## Author Contributions

**Conceptualization:** Daniel Faurholt-Jepsen, David L. Mwaniki, Michael K. Boit, Inge Tetens, Dirk L. Christensen.

**Data curation:** Dirk L. Christensen.

**Formal analysis:** Daniel Faurholt-Jepsen, Henrik Friis.

**Funding acquisition:** Dirk L. Christensen.

**Investigation:** Dirk L. Christensen.

**Methodology:** Daniel Faurholt-Jepsen, Henrik Friis, David L. Mwaniki, Michael K. Boit, Dirk L. Christensen.

**Project administration:** Dirk L. Christensen.

**Validation:** Daniel Faurholt-Jepsen, Henrik Friis, David L. Mwaniki, Michael K. Boit, Lydia U. Kaduka, Inge Tetens, Dirk L. Christensen.

**Visualization:** Daniel Faurholt-Jepsen, Henrik Friis, Dirk L. Christensen.

**Writing – original draft:** Dirk L. Christensen.

**Writing – review & editing:** Daniel Faurholt-Jepsen, Henrik Friis, David L. Mwaniki, Michael K. Boit, Lydia U. Kaduka, Inge Tetens.

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
