## [Decision Letter · Decision Letter 0]

4 Nov 2020

PONE-D-20-26651

Waist circumference and hypoalphalipoproteinemia as markers of cardiovascular risk in Kenyan adults

PLOS ONE

Dear Dr. Christensen,

Thank you for submitting your manuscript to PLOS ONE. After careful consideration, we feel that it has merit but does not fully meet PLOS ONE’s publication criteria as it currently stands. Therefore, we invite you to submit a revised version of the manuscript that addresses the points raised during the review process.

We look forward to receiving your revised manuscript.

Kind regards,

Giacomo Pucci

Academic Editor

PLOS ONE

Journal Requirements:

2. Please note that according to our data availability statement (https://journals.plos.org/plosone/s/data-availability), PLOS does not permit references to “data not shown.” Please provide the relevant data within the manuscript, the Supporting Information files, or in a public repository. If the data are not a core part of the research study being presented, please remove any references to these data. Thank you for your attention to this request.

5. Please include a copy of Tables 1 - 4 which you refer to in your text on page 17.

Reviewers' comments:

Reviewer's Responses to Questions

**Comments to the Author**

1. Is the manuscript technically sound, and do the data support the conclusions?

Reviewer #1: No

Reviewer #2: Yes

2. Has the statistical analysis been performed appropriately and rigorously? 

Reviewer #1: No

Reviewer #2: Yes

3. Have the authors made all data underlying the findings in their manuscript fully available?

Reviewer #1: Yes

Reviewer #2: Yes

4. Is the manuscript presented in an intelligible fashion and written in standard English?

Reviewer #1: Yes

Reviewer #2: Yes

5. Review Comments to the Author

Reviewer #1: General Comment - in this paper, Christensen et al. explore the effect of combination of HDL cholesterol values and waist circumference on other features of metabolic syndrome, such as fasting blood glucose, blood pressure, triglycerides, insulin reistance etc... in a large population of rural and urban Kenyans. The topic of increasing prevalence of metabolic syndrome in developing countries is relevant and hits my own interest. However, this paper does not add any relevant information on this topic. Basically, the main result of the paper is that the common definition of metabolic syndrome can be applied to Kenyans too, using appropriate cut-off levels for WC and HDLc (which were previously described). This is a quite trivial result. In my opinion, the main conceptual flaw of the paper is dealing WC and HDLc as independent factors, whereas their association is well estabilished since they are both criteria of metabolic syndrome. Similarly, the high probability of finding an additional feature of MetS in subjects with one feature of MetS, and even higher in subjects with two features of MetS, is a well estabilished notion and part of the definition of MetS itself (definitely not a peculiar feature of Kenyan population).

Title and Abstract - the endpoints of the study cannot be described as "cardiovascular risk" since they are not cardiovascular events, neither are they surrogate cardiovascular endpoints (such as carotid IMT, FMD, aortic stiffness, coronary calcium score, coronary stenosis etc...) Reported endpoints are instead metabolic parameters; so it would be more proper to describe them as "metabolic risk" or "cardiometabolic risk". This applies to the rest of the abstract and the text.

Results reported in the Abstract are too detailed. It is not clear what is the main finding of your research, whereas an abstract should clearly communicate this information.

Introduction - Introduction is accurate, but it does not explain the link between the background and the aim of the study. In other words: what is the unanswered question that you are addressing with your research?

Non-alcoholic fatty liver disease (NAFLD) is considered the liver manifestation of metabolic syndrome. I Authors agree to define their outcomes as "metabolic", rather than "cardiovascular", ther would be non necessity to define it as a separate analysis.

African populations share genetic determinants of HDL cholesterol values and HDL particles functionality (e.g. apolipoproteins L1). The impact of genetic variants should go beyond the prevalence of metabolic syndrome, as well as being correlated to other features of MetS. These aspects cannot be overcome in the introduction and should be at least briefly discussed in discussion.

Methods -

Since data about liver steatosis are reported, Authors should report if patients with other known causes of liver disease were excluded and if alcohol consumption was evaluated. If this was not, it would be a severe limitation for the validity of the study.

Page 8, Line 113 "HDL values were defined as <1.0 for men and <1.3 for women" Measure Units are missing.

Tables -

Table 1 should report also grades of liver steatosis.

Table 2 - data in columns should all be reported as beta (95%CI) or p-value or both.

Table 3 does not provide any additive information compared to Table 2

Reviewer #2: The manuscript by Faurholt-Jepsen et al. entitled: Waist circumference and hypoalphalipoproteinemia as markers of cardiovascular risk in Kenyan adults is well-written and brings forth important insight on the on WC – HDL relationship in the identification of CVD risk.

However, there are issues that need clarification:

1) The authors could consider changing the title from WC and “hypoalphalipoproteinemia” … to: WC and “low levels of high density lipoprotein-cholesterol”.

Technically the word “hypoalphalipoproteinemia” is correct. But throughout the manuscript they use “low HDL”.

2) The abstract should provide the age range. SD is already provided for age, but age range would be a very valuable addition. They should also add BMI with SD and range to the abstract. It is critical baseline data.

3) The authors do not present the data separately for men and women. They should consider this. Also their cohort has a mild predominance of women, they should state what they found in comparing pre and post menopausal women. Were the results the same? If they do not the menopausal status of the women, this should be stated as a limitation. But they could still arbitrarily divide the women into 2 groups (ie. above and below 48y). This is one reason why adding age range is so important.

4) They use hepatic steatosis as an end organ marker of cardiovascular risk. However, they need to recognize that this particular marker represents very advances disease in African descent populations. Hepatic steatosis is most well-studied in African Americans (AA). AA have much less hepatic steatosis than whites or Hispanics. And it is low levels of hepatic steatosis, which may explain why TG are low in the presence of insulin resistance. The authors are referred to: Guerrero, R. et al. Ethnic differences in hepatic steatosis: an insulin resistance paradox? Hepatology 2009; 49: 791-801.

In short, using hepatic steatosis as an end organ marker of CVD risk, will leave many people of African descent with CVD risk undiagnosed (just as using triglyceride does-triglyceride levels are closely linked to hepatic steatosis and VAT).

5) The methods section should describe how the participants were identified and recruited. It is not sufficient to refer to previous publications. In addition, they should add a limitation section. In the limitation section they need to state why they believe their sample is representative of Kenyans in general.

6) In the limitation section, they also need to acknowledge that their emphasis on high WC and low HDL represents a focus on insulin resistance as the cause of abnormal glucose tolerance and CVD. But in many African countries and low and middle income countries globally, abnormal glucose tolerance and the associated CVD is linked to relative beta-cell failure. The authors are referred to: Staimez L et al. Tale of Two Indians: Heterogeneity in Type 2 Diabetes Pathophysiology 2019; e3192.

7) Another limitation is the use of WC from the Ekoru et al article. The meta-analyses by Ekoru included studies which had people with diabetes in the prediction sample. WC is suppose to predict insulin resistance and who will get diabetes. So by including people with diabetes in the prediction of WC of risk, means people with diabetes are included in both the numerator and denominator. Further by including people with diabetes to predict the WC of risk-the results could be confounded: (a) uncontrolled or poorly controlled people with diabetes lose weight-and could confound the results by leading to spuriously low WC, (b) by including people with diabetes you are including people with diabetes due to beta-cell failure (which is associated with a lower BMI and a lower WC than when insulin resistance is the predominant cause) and again confounding results and leading to spuriously low WC.

Examples of studies done in Africa -which excluded people with diabetes and provide a much different picture and higher WC than the Ekoru study:

1) El Mabchour A, Delisle H, Vilgrain C, et al. Specific cut-off points for waist circumference and waist-to-height ratio as predictors of cardiometabolic risk in Black subjects: a cross-sectional study in Benin and Haiti. Diabetes Metab Syndr Obes 2015;8:513–23.

2) Peer N, Steyn K, Levitt N. Differential obesity indices identify the metabolic syndrome in Black men and women in Cape Town: the CRIBSA study. J Public Health 2016;38:175–82.

3) Prinsloo J, Malan L, de Ridder JH, et al. Determining the waist circumference cut off which best predicts the metabolic syndrome components in urban Africans: the SABPA study. Exp Clin Endocrinol Diabetes 2011;119:599–603.

So it is perfectly acceptable to use the Ekoru et al. But the controversy about the Ekoru study needs to be cited as a limitation.

6. PLOS authors have the option to publish the peer review history of their article (what does this mean?). If published, this will include your full peer review and any attached files.

Reviewer #1: **Yes: **Stefano Ministrini

Reviewer #2: No

---

## [Author Response · Author response to Decision Letter 0]

8 Jan 2021

Reviewer #1: General Comment - in this paper, Christensen et al. explore the effect of combination of HDL cholesterol values and waist circumference on other features of metabolic syndrome, such as fasting blood glucose, blood pressure, triglycerides, insulin reistance etc... in a large population of rural and urban Kenyans. The topic of increasing prevalence of metabolic syndrome in developing countries is relevant and hits my own interest. However, this paper does not add any relevant information on this topic. Basically, the main result of the paper is that the common definition of metabolic syndrome can be applied to Kenyans too, using appropriate cut-off levels for WC and HDLc (which were previously described). This is a quite trivial result. In my opinion, the main conceptual flaw of the paper is dealing WC and HDLc as independent factors, whereas their association is well estabilished since they are both criteria of metabolic syndrome. Similarly, the high probability of finding an additional feature of MetS in subjects with one feature of MetS, and even higher in subjects with two features of MetS, is a well estabilished notion and part of the definition of MetS itself (definitely not a peculiar feature of Kenyan population).

RESPONSE: We are building our study on the early attempt by Després and colleagues and later on Sam and colleagues in DM patients to identify two simple clinical parameters for detecting (increased) risk of CVD/CMD. They used central obesity in combination with triglycerides. However, as WHO standard cut-off for central obesity may not fit African populations, and elevated triglycerides are uncommon as a dyslipidemic feature in African individuals, we combined central obesity based on newly published central obesity cut-offs for increased CVD/CMD risk, and the common low HDL cholesterol dyslipidemic feature in Africans. We agree that some of the data we have presented are MetS features. However, this does not apply for VAT and SAT measurements as well as for liver steatosis.

Title and Abstract - the endpoints of the study cannot be described as "cardiovascular risk" since they are not cardiovascular events, neither are they surrogate cardiovascular endpoints (such as carotid IMT, FMD, aortic stiffness, coronary calcium score, coronary stenosis etc...) Reported endpoints are instead metabolic parameters; so it would be more proper to describe them as "metabolic risk" or "cardiometabolic risk". This applies to the rest of the abstract and the text.

RESPONSE: We have changed “cardiovascular risk” to “cardiometabolic risk” in the title, abstract, key words, and main text (where appropriate).

Results reported in the Abstract are too detailed. It is not clear what is the main finding of your research, whereas an abstract should clearly communicate this information.

RESPONSE: We have shortened the Abstract accordingly – it turned out to be too long to begin with.

Introduction - Introduction is accurate, but it does not explain the link between the background and the aim of the study. In other words: what is the unanswered question that you are addressing with your research?

RESPONSE: Apart from adding/changing a few words, we have added the following sentence to make the aim of the study more clear:

“Thus, replacing triglyceride with HDL in combination with lower WC cut-off than used by Deprés and colleagues seems more appropriate as cardiometabolic risk markers in black, African populations”.

Non-alcoholic fatty liver disease (NAFLD) is considered the liver manifestation of metabolic syndrome. I Authors agree to define their outcomes as "metabolic", rather than "cardiovascular", ther would be non necessity to define it as a separate analysis.

RESPONSE: We agree that NAFLD is a result of hyperglycaemia, and thereby MetS. We have therefore removed NAFLD from the aim of the study. However, we have kept the analysis in order to show the proportion of NAFLD.

African populations share genetic determinants of HDL cholesterol values and HDL particles functionality (e.g. apolipoproteins L1). The impact of genetic variants should go beyond the prevalence of metabolic syndrome, as well as being correlated to other features of MetS. These aspects cannot be overcome in the introduction and should be at least briefly discussed in discussion.

RESPONSE: We acknowledge that APOL1 allele frequencies (G1 and G2 variants) are important in an African context. However, in the current context, they are primarily related to non-diabetic kidney disease, and we believe it is not possible to briefly discuss this genetic determinant in the context of HDL values in a meaningful way. It would require an introduction to the HDL cholesterol and APOL1 relationship before referring to the studies on for example CKD. Thus, we have decided not to include the proposed genetic aspects in the discussion.

Methods -

Since data about liver steatosis are reported, Authors should report if patients with other known causes of liver disease were excluded and if alcohol consumption was evaluated. If this was not, it would be a severe limitation for the validity of the study.

RESPONSE: We have identified the few individuals (n=26) who qualify for AFLD due to their relatively high alcohol consumption. However, excluding these from the analyses make no substantial difference, and therefore we have kept them in the analyses:

“We identified 26 individuals with excessive, self-reported alcohol consumption that exceeded criteria for AFLD (16). Excluding these individuals from the hepatic steatosis analyses did not alter the results (not shown)”.

We did not measure any other causes of possible liver disease, and have therefore added the following sentence as a study limitation:

“We acknowledge several limitations to this study. First, we did not test for hepatitis B or C which is a limitation to our hepatic measurements”. 

Page 8, Line 113 "HDL values were defined as <1.0 for men and <1.3 for women" Measure Units are missing.

RESPONSE: We have added mmol/L as units behind the HDL values.

Tables -

Table 1 should report also grades of liver steatosis.

RESPONSE: We have added three grades of hepatic steatosis: mild, moderate, and severe.

Table 2 - data in columns should all be reported as beta (95%CI) or p-value or both.

RESPONSE: All data in columns are already reported as beta (95% CI).

Table 3 does not provide any additive information compared to Table 2

RESPONSE: We prefer to keep it, as the results are based on cut-points used in clinical practice. However, if the reviewer insists, we will remove Table 3

Reviewer #2: The manuscript by Faurholt-Jepsen et al. entitled: Waist circumference and hypoalphalipoproteinemia as markers of cardiovascular risk in Kenyan adults is well-written and brings forth important insight on the on WC – HDL relationship in the identification of CVD risk.

REPONSE: We thank the Reviewer for the positive comments.

However, there are issues that need clarification:

1) The authors could consider changing the title from WC and “hypoalphalipoproteinemia” … to: WC and “low levels of high density lipoprotein-cholesterol”. Technically the word “hypoalphalipoproteinemia” is correct. But throughout the manuscript they use “low HDL”.

RESPONSE: We have followed the suggestion by the Reviewer and changed “hypoalphalipoproteinemia” to “low high-density lipoprotein cholesterol” in the title of manuscript.

2) The abstract should provide the age range. SD is already provided for age, but age range would be a very valuable addition. They should also add BMI with SD and range to the abstract. It is critical baseline data.

RESPONSE: Even though this would be in conflict with the comments of Reviewer#1 who has asked us to remove results from the Abstract, we have decided to comply with these comments and added the requested information. However, we have decided to replace SD with range as both results are not essential in the Abstract in our view, and it would require removal of other information due to word restrictions.

3) The authors do not present the data separately for men and women. They should consider this. Also their cohort has a mild predominance of women, they should state what they found in comparing pre and post menopausal women. Were the results the same? If they do not the menopausal status of the women, this should be stated as a limitation. But they could still arbitrarily divide the women into 2 groups (ie. above and below 48y). This is one reason why adding age range is so important.

RESPONSE: We have already included several stratifications, and we have therefore chosen to adjust instead of stratifying for sex. We do not have information on pre- or postmenopausal status in the women, and have mentioned this as a limitation. We have adjusted rather than stratified for age, and the results were the same as in unadjusted analyses of the estimates. Thus, the interpretation of the results would not change. In brief, sex and age do not change the results. 

4) They use hepatic steatosis as an end organ marker of cardiovascular risk. However, they need to recognize that this particular marker represents very advances disease in African descent populations. Hepatic steatosis is most well-studied in African Americans (AA). AA have much less hepatic steatosis than whites or Hispanics. And it is low levels of hepatic steatosis, which may explain why TG are low in the presence of insulin resistance. The authors are referred to: Guerrero, R. et al. Ethnic differences in hepatic steatosis: an insulin resistance paradox? Hepatology 2009; 49: 791-801.

In short, using hepatic steatosis as an end organ marker of CVD risk, will leave many people of African descent with CVD risk undiagnosed (just as using triglyceride does-triglyceride levels are closely linked to hepatic steatosis and VAT).

RESPONSE: We thank the reviewer for adding the reference. While we agree that there is a paucity hepatic steatosis data in black African populations, and therefore we currently need to lean on data in African American populations, it important to acknowledge that using African Americans as a surrogate black African population is not without problems/limitations due to genetic admixture. It is also of note that the vast majority of individuals in our study with NAFLD had mild steatosis. Nevertheless, we have added the concerns raised here by the Reviewer as a limitation: 

“Another limitation concerning the hepatic steatosis results we need to emphasize is the relatively low proportion (15.3 % of which 93.1 % was mild steatosis, and none had severe steatosis) we found in a sub-group of the study population. This could partly be explained by the low triglyceride levels seen in African populations, and such an association has been reported in African Americans compared to white and Hispanic US populations (22)”.

5) The methods section should describe how the participants were identified and recruited. It is not sufficient to refer to previous publications. In addition, they should add a limitation section. In the limitation section they need to state why they believe their sample is representative of Kenyans in general.

RESPONSE: We have added the requested information in the main text including a separate limitation section.

6) In the limitation section, they also need to acknowledge that their emphasis on high WC and low HDL represents a focus on insulin resistance as the cause of abnormal glucose tolerance and CVD. But in many African countries and low and middle income countries globally, abnormal glucose tolerance and the associated CVD is linked to relative beta-cell failure. The authors are referred to: Staimez L et al. Tale of Two Indians: Heterogeneity in Type 2 Diabetes Pathophysiology 2019; e3192.

RESPONSE: We acknowledge this potential limitation to our study results, even though IFG – a marker of beta-cell failure as the predominant factor for pre-DM – was very low in the current study population as opposed to IGT – where (peripheral) insulin resistance is the predominant factor for pre-DM – which was relatively high. Accordingly, we have added a sentence to the Limitations section. 

7) Another limitation is the use of WC from the Ekoru et al article. The meta-analyses by Ekoru included studies which had people with diabetes in the prediction sample. WC is suppose to predict insulin resistance and who will get diabetes. So by including people with diabetes in the prediction of WC of risk, means people with diabetes are included in both the numerator and denominator. Further by including people with diabetes to predict the WC of risk-the results could be confounded: (a) uncontrolled or poorly controlled people with diabetes lose weight-and could confound the results by leading to spuriously low WC, (b) by including people with diabetes you are including people with diabetes due to beta-cell failure (which is associated with a lower BMI and a lower WC than when insulin resistance is the predominant cause) and again confounding results and leading to spuriously low WC.

Examples of studies done in Africa -which excluded people with diabetes and provide a much different picture and higher WC than the Ekoru study:

1) El Mabchour A, Delisle H, Vilgrain C, et al. Specific cut-off points for waist circumference and waist-to-height ratio as predictors of cardiometabolic risk in Black subjects: a cross-sectional study in Benin and Haiti. Diabetes Metab Syndr Obes 2015;8:513–23.

2) Peer N, Steyn K, Levitt N. Differential obesity indices identify the metabolic syndrome in Black men and women in Cape Town: the CRIBSA study. J Public Health 2016;38:175–82.

3) Prinsloo J, Malan L, de Ridder JH, et al. Determining the waist circumference cut off which best predicts the metabolic syndrome components in urban Africans: the SABPA study. Exp Clin Endocrinol Diabetes 2011;119:599–603.

So it is perfectly acceptable to use the Ekoru et al. But the controversy about the Ekoru study needs to be cited as a limitation.

RESPONSE: We agree, and the cut-points are still up for debate. We decided to include those with diabetes as it is a continuum of dysglycaemia. We also ran the analyses with and without those on treatment (n=22), which did not affect the results. Thus, we prefer to keep the participants in the model. If the reviewer insists, we will revise the data and leave them out.

---

## [Decision Letter · Decision Letter 1]

2 Feb 2021

PONE-D-20-26651R1

Waist circumference and low high-density lipoprotein cholesterol as markers of cardiometabolic risk in Kenyan adults

PLOS ONE

Dear Dr. Christensen,

Thank you for submitting your manuscript to PLOS ONE. The reviewers have considered positively your changes made on the original manuscript. However, there is still a question raised by reviewer #2 that needs to be solved. Therefore, we invite you to submit a revised version of the manuscript that addresses this specifical point. 

We look forward to receiving your revised manuscript.

Kind regards,

Giacomo Pucci

Academic Editor

PLOS ONE

Reviewers' comments:

Reviewer's Responses to Questions

**Comments to the Author**

1. If the authors have adequately addressed your comments raised in a previous round of review and you feel that this manuscript is now acceptable for publication, you may indicate that here to bypass the “Comments to the Author” section, enter your conflict of interest statement in the “Confidential to Editor” section, and submit your "Accept" recommendation.

Reviewer #1: All comments have been addressed

Reviewer #2: (No Response)

2. Is the manuscript technically sound, and do the data support the conclusions?

Reviewer #1: Yes

Reviewer #2: Yes

3. Has the statistical analysis been performed appropriately and rigorously? 

Reviewer #1: Yes

Reviewer #2: Yes

4. Have the authors made all data underlying the findings in their manuscript fully available?

Reviewer #1: No

Reviewer #2: Yes

5. Is the manuscript presented in an intelligible fashion and written in standard English?

Reviewer #1: Yes

Reviewer #2: Yes

6. Review Comments to the Author

Reviewer #1: My comments have been adequately addressed. I'm still concerned about the limited scientific and practical relevance of the results.

Reviewer #2: There is only one necessary change. For BMI and age both SD and range have to be included.

SD gives an idea of skewness of data and does not offset the need for range and vice aversa.

The Reviewer has never seen an occasion where range was allowed eliminate the need for range.

7. PLOS authors have the option to publish the peer review history of their article (what does this mean?). If published, this will include your full peer review and any attached files.

Reviewer #1: **Yes: **Stefano MInistrini

Reviewer #2: No

---

## [Author Response · Author response to Decision Letter 1]

4 Feb 2021

Reviewer #2: There is only one necessary change. For BMI and age both SD and range have to be included.

SD gives an idea of skewness of data and does not offset the need for range and vice aversa.

The Reviewer has never seen an occasion where range was allowed eliminate the need for range.

Response: Point taken and SD has been included for BMI. In order to ensure uniform reporting of results in the Abstract section, we have also added SD for age.

---

## [Editor Report · Decision Letter 2]

10 Feb 2021

Waist circumference and low high-density lipoprotein cholesterol as markers of cardiometabolic risk in Kenyan adults

PONE-D-20-26651R2

Dear Dr. Christensen,

We’re pleased to inform you that your manuscript has been judged scientifically suitable for publication and will be formally accepted for publication once it meets all outstanding technical requirements.

Kind regards,

Giacomo Pucci

Academic Editor

PLOS ONE

---

## [Editor Report · Acceptance letter]

17 Feb 2021

PONE-D-20-26651R2 

Waist circumference and low high-density lipoprotein cholesterol as markers of cardiometabolic risk in Kenyan adults 

Dear Dr. Christensen:

I'm pleased to inform you that your manuscript has been deemed suitable for publication in PLOS ONE. Congratulations! Your manuscript is now with our production department. 

Kind regards, 

on behalf of

Dr. Giacomo Pucci 

Academic Editor

PLOS ONE